# Motility, Adhesion and c-di-GMP Influence the Endophytic Colonization of Rice by *Azoarcus* sp. CIB

**DOI:** 10.3390/microorganisms9030554

**Published:** 2021-03-08

**Authors:** Helga Fernández-Llamosas, Eduardo Díaz, Manuel Carmona

**Affiliations:** Microbial and Plant Biotechnology Department, Centro de Investigaciones Biológicas Margarita Salas—CSIC, Ramiro de Maeztu 9, 28040 Madrid, Spain; hfllamosas@gmail.com (H.F.-L.); ediaz@cib.csic.es (E.D.)

**Keywords:** flagella, *pili* type IV, c-di-GMP, *Azoarcus*, endophyte, plant bacteria interaction

## Abstract

Proficient crop production is needed to ensure the feeding of a growing global population. The association of bacteria with plants plays an important role in the health state of the plants contributing to the increase of agricultural production. Endophytic bacteria are ubiquitous in most plant species providing, in most cases, plant promotion properties. However, the knowledge on the genetic determinants involved in the colonization of plants by endophytic bacteria is still poorly understood. In this work we have used a genetic approach based on the construction of *fliM*, *pilX* and *eps* knockout mutants to show that the motility mediated by a functional flagellum and the *pili* type IV, and the adhesion modulated by exopolysaccarides are required for the efficient colonization of rice roots by the endophyte *Azoarcus* sp. CIB. Moreover, we have demonstrated that expression of an exogenous diguanylate cyclase or phophodiesterase, which causes either an increase or decrease of the intracellular levels of the second messenger cyclic di-GMP (c-di-GMP), respectively, leads to a reduction of the ability of *Azoarcus* sp. CIB to colonize rice plants. Here we present results demonstrating the unprecedented role of the universal second messenger cyclic-di-GMP in plant colonization by an endophytic bacterium, *Azoarcus* sp. CIB. These studies pave the way to further strategies to modulate the interaction of endophytes with their target plant hosts.

## 1. Introduction

Plant associated bacteria play a fundamental role in the healthy state of the plant by contributing to its nutrition, development and defense [1,2,3]. A good number of the plant-associated bacteria colonize the plant tissues as endophytes being the roots the major target for endophytic colonization [2]. The endophytes are microorganisms that spend most of their life cycle inside plant tissues without causing symptoms of plant damage [4], but some endophytes are able to live outside of the plant without losing the capacity to colonize seedlings [5]. In addition, some endophytic bacteria exhibit beneficial effects on the host plant, such as plant growth promotion, the induction of increased resistance to pathogens, as well as the supply of fixed nitrogen to the host plant [6]. The effective root colonization by endophytic bacteria involves the coordinated expression of a number of gene factors, many of which are still poorly understood [7,8,9,10]. To analyze the genes involved in plant colonization several genomes from endophytic bacteria were compared [2,11,12,13,14]. Moreover, to establish the determinants that differentiate endophytes from non-endophytic bacteria able to interact with plants, genomes from endophytic bacteria were compared with those from bacterial plant pathogens, bacteria that colonize the rhizosphere and bacteria from the rhizobia group [12]. Some properties such as motility, chemotaxis and adherence were considered crucial for initiating endophytic colonization. Bacterial motility plays an important role in plant colonization. Since endophytes are attracted to the root by plant exudates [2], most of them have flagella [14]. Chemotaxis towards the plant varies greatly between the different species of endophytes according to the rank of their preferred host [11]. For example, while in *Gluconacetobacter diazotrophicus* only 12 genes related to transmembrane receptors, and two-component response regulators are involved in the signal transduction for chemotaxis, the genome of *Azoarcus* sp. BH72 has 51 such genes, a number similar to that of *Burkholderia phytofirmans* PsJN [14].

The flagellum is considered one of the first bacterial factors that come into contact with the plant, activating the induced systemic resistance [15]. It has been reported that the *Azoarcus* sp. BH72 flagellum has an active role during the effective endophytic colonization of rice roots [16]. Once the bacteria contacted with plant roots, adherence and motility of the twitching type mediated by the *pili* type IV becomes especially important [11,17]. Thus, the presence of functional *pili* seems to be also essential for the colonization of rice by *Azoarcus* sp. BH72 [7,18]. After the adhesion of the bacteria to the roots microcolonies are formed involving the essential participation of exopolysaccharides and lipopolysaccharides [19,20,21], as it has been reported in the colonization of rice roots by *Herbaspirillum seropedicae* [22] and *G. diazotrophicus* [23]. In fact, bacterial exopolysaccharide is related with a multitude of vital functions for the cell, supporting the cohesion and adhesion to biotic and abiotic surfaces through cell aggregation and participating in biofilm formation [24]. In fact, some reports have suggested the participation of exopolysaccharide in the establishment of symbiotic and plant-bacterial interactions between rhizobia and legumes [25], and in the rhizosphere interactions of some strains of *P. fluorescens* [26] or endophytes such *G. diazotrophicus* [23].

Cyclic di-GMP (c-di-GMP) is a second universal messenger in prokaryotes that controls many bacterial processes such as the transition from a planktonic to a sessile state favoring the establishment of biofilms and modulating bacterial virulence [27,28]. In fact, it has been reported that the levels of c-di-GMP in bacteria control the synthesis of exopolysaccharides, adhesins and the formation of biofilms [29,30,31,32,33], hence regulating, among others, colony morphology [34,35], quorum sensing processes [36], cell motility [37,38,39,40], nodulation [33,41] and virulence [42,43]. However, reports on the role of c-di-GMP in plant-bacteria interactions are scare. The few available studies have mainly focused on the regulation of virulence in plant pathogenic bacteria [44] such as *Pseudomonas syringae* [45,46], *P. savastanoi* [45], *Xanthomonas campestris* [47,48,49] or *X. oryzae* [50,51], and in the study of the interaction between plants and symbiotic bacteria [45,52]. The results obtained from these studies show disparity. Increasing c-di-GMP levels by the overexpression of the diguanylate cyclase (DGC) PleD in pathogenic strains *P. syringae* pv. *tomato* and *P. syringae* pv. *phaseolicola* reduces the expression of the type III secretion system (associated with pathogenesis) without affecting virulence. However, the overexpression of PleD in *P. savastanoi* does not affect the expression of the secretion system but delays the appearance of necrosis [45]. In general, high concentrations of cyclic di-GMP promote biofilm formation, aggregation, attachment, and reduced motility of pathogens such as *Vibrio cholerae* [53], *Xanthomonas campestris* pv. *campestris* [54] or *Yersinia pestis* [55]. In plant associated bacteria, the study of how c-di-GMP levels affect interactions, for example between *Rhizobium etli* and *Phaseolus vulgaris* or between *R. leguminosarum* and *Vicia sativa*, have shown that, in both cases, high levels of c-di-GMP favor adhesion to the roots of the plant, associated with a greater formation of biofilm, but negatively affect later stages, reducing nitrogen fixation and, consequently, plant growth [45]. Nevertheless, and as far as we know, there are no reports on the role of c-di-GMP in plant colonization by endophytes.

*Azoarcus* sp. CIB is a beta-proteobacterium, facultative anaerobic (denitrifying) strain able to colonize rice roots as an endophyte [56] that display plant growth promoting traits such as the ability to fix nitrogen, solubilize inorganic phosphate, produce the plant growth hormone indoleacetic acid [56] and promote the rice plant growth under certain environmental stress conditions [57]. Some genetic determinants likely involved in the endophytic interaction, such as the presence of flagella and *pili* type IV, can be inferred from the genome sequence of strain CIB [58]. In addition, strain CIB owns some other interesting biotechnological properties such as the ability to degrade toxic aromatic compounds, e.g., toluene/xylene, under aerobic and anaerobic conditions [59,60], and tolerate high concentrations of certain metals and metalloids for their conversion into metal nanoparticles of industrial value [61,62]. All these features make strain CIB significantly different to the typical *Azoarcus* endophytic strains, and more closely related to members of the new genus *Aromatoleum* [63].

The aim of this work is to explore the role of the flagellum, *pili* type IV and exopolysaccarides on the colonization of rice roots by *Azoarcus* sp. CIB. Since the second messenger c-di-GMP is connected with motility and adhesion, in this work we investigated if the levels of c-di-GMP also control the ability of the bacterium to colonize plant, an aspect scarcely explored in endophytes.

## 2. Materials and Methods

### 2.1. Strains, Seeds and Plasmids Used

The bacterial strains and plasmids used in this work are listed in Table 1. *Azoarcus* sp. strain CIB was deposited in the Spanish Type Culture Collection (CECT #5669). Seeds from *Oryza sativa* L. GLEVA were kindly provided by Castells Seeds Co. (Tarragona, Spain).

### 2.2. Bacterial Growth Conditions

*Azoarcus* strains were grown on MC medium (MA basal medium plus trace elements and vitamins). MA basal medium is detailed in Table 2 [59]. As carbon source, 0.2% (*w/v*) pyruvate was added. When needed, bacterial strains were also grown on a variant of the VM-ethanol rich medium [70] with the composition detailed in Table 2. *E. coli* strains were grown in lysogeny broth (LB) medium [71] at 37 °C. When required, kanamycin (Km) or gentamicin (Gm) was added at 50 μg mL^−1^ or 10 μg mL^−1^, respectively. The growth of the cultures was monitored by measuring the absorbance at 600 nm (*A*_600_) using a Shimadzu UV-260 spectrophotometer or by counting viable cells. Cell morphology was analyzed with a Nikon OPTIPHOT-2 phase contrast microscope.

### 2.3. Molecular Biology Techniques

Standard molecular biology techniques were performed as previously described [72]. DNA fragments were purified with Gene-Turbo (BIO101 Systems). Plasmids and PCR products were purified with a High Pure Plasmid and PCR Product Purifications kits (Roche), respectively. Oligonucleotides were supplied by Sigma Co and they are detailed in Appendix A. All cloned inserts and DNA fragments were confirmed by DNA sequencing with fluorescently labeled dideoxynucleotide terminators [73] and AmpliTaq FS DNA polymerase (Applied Biosystems) in an ABI Prism 377 automated DNA sequencer (Applied Biosystems, Foster City, CA, USA). Transformations of *E. coli* were carried out by using the RbCl method or by electroporation (Gene Pulser, Bio-Rad, Hercules, CA, USA) [72]. Transformation of *Azoarcus* sp. CIB was done by biparental conjugation using the strain *E. coli* S17-1λ*pir* as donor, following a protocol previously established [64] with slight modifications: donor cells were grown to *A*_600_ of 5 and the receptor CIB strain was grown on MC medium supplemented with pyruvate 0.2%, and concentrated to reach an *A*_600_ of 35. The transconjugants were selected on MC medium supplemented with 10 mM glutarate plus the corresponding antibiotic.

### 2.4. Construction of Azoarcus sp. CIBdpilX, Azoarcus sp. CIBdfliM and Azoarcus sp. CIBdepsF Mutant Strains

For insertional disruption of *pilX*, *fliM* and *epsF* genes in the genome of *Azoarcus* sp. CIB we used single homologous recombination, that usually generates polar effects downstream of the the mutated gene, with PCR-amplified DNA fragments obtained with the primer pairs 5′pilX/3′pilX; 5′fliM/3′fliM and 5′epsF/3′epsF, respectively (Appendix A). The obtained fragments were double-digested with the appropriate restriction enzymes, generating the pK18mobpilX, pK18mobfliM and pK18mobepsF recombinant plasmids (Table 1). These plasmids were transferred from *E. coli* S17-1λ*pir* (donor strain) to *Azoarcus* sp. CIB (recipient strain) by biparental filter mating [72], and exconjugants strains *Azoarcus* sp. CIBd*pilX*, *Azoarcus* sp. CIBd*fliM* and *Azoarcus* sp. CIBd*epsF* were isolated on kanamycin-containing MC agar plates harboring 10 mM glutarate as the sole carbon source for counterselection of donor cells. The mutant strains were verified by PCR amplification using a standard oligonucleotide (F24 or R24) flanking the multiple cloning site of the plasmid and a second specific oligonucleotide external to the fragment used to perform the homologous recombination (Appendix A). RT-PCRs were used to ensure that the mutated genes were not expressed in their corresponding host mutant strains (Appendix A).

### 2.5. RNA Extraction and Quantitative Reverse Transcription-PCR (qRT-PCR) Experiments

RNA was purified from bacterial cells grown up to the middle of the exponential phase and resuspended in a solution containing TE buffer (Tris-HCl 10 mM pH 8.0, EDTA 1 mM) and lysozyme 50 mg mL^−1^ (Sigma). Total RNA was obtained using High Pure RNA Isolation Kit (Roche). The DNA was removed with DNAse and Removal Treatment Kit (Ambion). The concentration and purity of RNA was spectrophotometrically determined at *A*_260_ and calculating the *A*_260_/*A*_280_ ratio, respectively. The cDNA was obtained by using the Transcriptor First Strand cDNA Synthesis kit (Roche). Each RT reaction (20 μL) contained 1 μg RNA, 10 U reverse transcriptase, RNAse inhibitor 20 U, dNTPS 1 mM and 60 μM random hexamer primers. The standard procedure of cDNA production includes a 10 min incubation at 25 °C followed by 1 cycle of 30 min at 55 °C and another incubation of 5 min at 85 °C using the Mastercycler Gradient equipment (Eppendorf). Then, 1 μL of the obtained cDNA was used as template for the PCR. The analysis was performed in three technical replicates from three biological samples. Reactions (20 μL) contained 1 μL of cDNA, 0.25 μM of each of the three target-specific primers, and 10 μL of SYBR Green I Master Mix (Roche). Oligonucleotides 5′RTpilY1 and 3′RTpilY1 (Appendix A) were used to amplify transcripts from the *pilY1* gene. Oligonucleotides 5′RTfliC and 3′RTfliC (Appendix A) were used to amplify transcripts from the *fliC* gene. Oligonucleotides 5′RTepsF and 3′RTepsF (Appendix A) were used to amplify transcripts from the *epsF* gene. Oligonucleotides 5′POLIIIHK and 3′POLIIIHK (Appendix A) were used to amplify transcripts from the *dnaE* gene (αDNApol III subunit gene) used as an internal control to normalize the sample data as reported before [74]. PCR amplifications were carried out with one denaturation cycle (95 °C for 5 min), followed by 30 cycles of amplification (95 °C for 10 s, 60 °C for 10 s, and 72 °C for 10 s). After amplification, 7 melting curves were generated to confirm amplification of a single product. For relative quantification of the fluorescence values, a calibration curve was constructed by seven-fold serial dilutions of *Azoarcus* sp. CIB genomic DNA sample ranging from 1 ng to 0.5 × 10^−7^ ng. This curve was then used as a reference standard for extrapolating the relative abundance of the cDNA target within the linear range of the curve. The results are shown as relative quantification using the ΔΔCt method [75]

### 2.6. Inoculation of Rice Seedlings with Bacteria

Dehulled rice seeds (*O. sativa* L. GLEVA) were surface sterilized by shaking them at 25 °C for 30 min in 30 mL 1% (*v/v*) sodium hypochlorite. After rinsing them three times for 10 min in sterile water, the seeds were incubated in VM-ethanol for 48 h. The germ-free seedlings were selected for inoculation. The germination of the seeds continued on humidified filter papers for 24 h prior to inoculation with bacterial cells. Bacteria were obtained as described previously [56]. After growth up to mid-exponential phase, bacterial cells were collected by centrifugation, washed with sterile 0.9% NaCl (*w/v*) solution, resuspended in 1 mL of sterile distilled water, and the cell suspension was inoculated onto the surface of each seedling in aseptic conditions. After inoculation, the seedlings were grown at 25 °C under natural daylight conditions (10 h of light and 14 h of darkness) for 5–10 days.

### 2.7. Recovery and Quantification of Endophytes

Rice seedlings were collected 5 days after inoculation with bacterial strains. The roots were weighed, proceeding to its surface disinfection by immersion for 3 min in a 1% (*v/v*) solution of sodium hypochlorite and three washes with sterile water. The roots were then homogenized by adding 1 mL of saline (0.85% NaCl) solution to a Potter-Elvehjem [76]. Once the extract was obtained, serial dilutions were made and plated in VM medium plates (1.5% agar) supplemented with kanamycin in order to determine the number of Colony Forming Units (CFUs). Each experiment was performed by triplicate (*n* = 3) and the total number of roots collected for each replica was 20.

### 2.8. Motility Tests

Swimming-type motility assays were performed on semi-solid VM-agar 0.3% medium plates using cultures grown in VM-ethanol liquid medium up to an *A*_600_ of 0.6. For swarming motility assays, plates of MC medium with 0.2% pyruvate (*w/v*) and semi-solid 0.4% agar were used employing cultures grown in liquid MC medium with 0.2% pyruvate (*w/v*) up to an *A*_600_ of 0.6. The plates were incubated for 7 days at 30 °C and the diameter of the growth halos of the different *Azoarcus* sp. CIB strains was measured.

### 2.9. Colony Morphology

The morphological study of the colonies of *Azoarcus* sp. CIB strains was performed on VM medium plates supplemented with 0.5 g L^−1^ of Congo Red dye (Sigma-Aldrich, San Louis, MO, USA). The bacteria were grown in VM-ethanol liquid medium up to an *A*_600_ of 0.6. Subsequently, serial dilutions were made in saline solution (0.85% NaCl) and plated on VM-Congo Red plates. The plates were incubated for 7 days at 30 °C and the morphology of isolated colonies was analyzed with a Leica MZ16FA stereomicroscope.

### 2.10. Sequence Data Analyses

Nucleotide sequence analyses were done at the National Center for Biotechnology Information (NCBI) server (https://www.ncbi.nlm.nih.gov/, accessed on 7 March 2021).

### 2.11. Statistical Analysis

The data were analyzed with the GraphPad software package (San Diego, CA, USA) using paired or unpaired *t*-test. Statistical differences were presented as *p <* 0.01 (**) or *p <* 0.05 (*).

## 3. Results and Discussion

### 3.1. A Functional Flagellum and Pili Type IV Are Required for Rice Colonization of Azoarcus sp. CIB

As mentioned in the Introduction, bacterial motility plays an important role in plant colonization. The flagellum has traditionally been regarded as the main responsible for bacterial motility. Three gene clusters predicted to control the synthesis and regulation of the flagellum have been identified at three different positions of the genome of *Azoarcus* sp. CIB ([58], Appendix A). We then checked whether a mutation that abolish flagellum functionality may have an effect on the endophytic colonization of rice by strain CIB.

To try to generate a motility minus phenotype in *Azoarcus* sp. CIB, we constructed a mutant strain in the *fli* operon. The *Azoarcus* sp. CIBd*fliM* strain (Table 1) harbors a disruptional insertion of the *fliM* (AzCIB_0962) gene, encoding the FliM protein of the C ring which has been shown to be directly involved in flagellum movement and the flagellar export apparatus [77,78,79,80]. To check whether the *fliM* mutant strain showed an altered motility phenotype, we performed bacterial swimming assays. As shown in Figure 1A, the movement of the mutant cells was significantly diminished (around 50%) with respect to that of the parental strain, hence suggesting an important loss of functionality of the flagellum in *Azoarcus* sp. CIBd*fliM* strain.

We then checked the ability of the mutant strain to colonize rice seedlings. Interestingly, the number of endophytes recovered after inoculation of rice seedlings with *Azoarcus* sp. CIBd*fliM* strain was around an order of magnitude lower (3.4 × 10^3^ per g of roots) than those recovered from seedlings inoculated with the wild type CIB strain (4.5 × 10^4^ per g of roots) (Figure 1B). These results indicate an important role of the flagellum at some stage of the endophytic colonization of rice by *Azoarcus* sp. CIB. These results are in agreement with previous reports showing that deletion of the *fliC3* gene, encoding the main flagellin (FliC) of the flagellum, in the endophyte *Azoarcus* sp. BH72 strain leads to a significant decrease of around 80% in the plant colonizing capacity on competition assays between the wild type and the mutant strain [16], hence supporting that the flagellum is involved in the *Azoarcus*-plant interaction.

Different studies conducted on *Azoarcus* spp. strains have concluded that *pili* type IV are needed for plant colonization since they are responsible of adhesion and movement to the plant roots [7,16,18,81]. The genes predicted to be responsible for the synthesis of *pili* type IV in *Azoarcus* sp. CIB are organized in several gene clusters distributed along the genome ([58], Appendix A). In order to analyze if the *pil* genes from *Azoarcus* sp. CIB are involved in the colonization of rice root, we constructed an *Azoarcus* sp. CIBd*pilX* mutant strain containing a disruption of the *pilX* gene (AzCIB_3117) (Table 1). The motility (swarming pattern) of the *pilX* mutant strain was reduced by 40% compared to that observed with the wild-type strain (Figure 2A), which is in agreement with the already observed lower motility in *pil* mutant bacterial strains [82,83] and confirms that the *pil* genes indeed encode proteins that participate in the motility apparatus in *Azoarcus* sp. CIB.

Interestingly, the plant colonization ability of the *Azoarcus* sp. CIBd*pilX* strain was more than one order of magnitude lower than that of the parental CIB strain (Figure 2B). Therefore, these results allowed us to conclude that *pili* type IV are involved in the interaction of the CIB strain with rice. Our results in strain CIB are in agreement with those reported in *Azoarcus* sp. BH72 where a deletion of *pilX* did not inhibit the synthesis of the *pili* but affected its functionality [81,82,83,84,85] and decreased by 80% the plant colonizing capacity of the mutant strain [7,18].

In summary, all these results taken together indicate that both, flagellum and *pili* type IV are needed for the interaction of *Azoarcus* sp. CIB with rice.

### 3.2. Role of the Genes Involved in Exopolysaccharide Modification in the Interaction between Azoarcus sp. CIB and Rice

As mentioned in the Introduction, exopolysaccharides have an important role in plant-bacteria interactions [23,25,26]. In the genome of *Azoarcus* sp. CIB we have identified three gene clusters putatively involved in the synthesis and modification of exopolysaccharides ([56], Appendix A). Previous results have shown that the gene cluster *AzCIB_0813-0833* was strongly induced at the transcriptional level in the presence of rice extracts [57]. To analyze the potential role of this gene cluster in the colonization of rice roots by the CIB strain, it was inactivated by insertional disruption at the *AzCIB_0818* (*epsF*) gene. The surface of the colonies of the *Azoarcus* sp. CIBd*epsF* mutant strain grown on a solid culture medium supplemented with Congo Red dye [86] showed a rugose phenotype in contrast to the smooth phenotype shown by colonies of the wild-type strain (Figure 3A). These changes in colony morphology strongly suggest the existence of significant alterations in the exopolysaccharide of the *epsF* mutant strain [87], hence indicating that the *eps* genes are indeed involved in exopolysaccharide synthesis/modification. The next step was to study the ability of the strain *Azoarcus* sp. CIBd*epsF* to colonize rice roots as an endophyte. The plant colonization results showed that the number of *Azoarcus* sp. CIBd*epsF* endophytes recovered from rice roots (1.65 × 10^4^ bacteria per g of roots) was 36% lower than that obtained from roots inoculated with the parental strain *Azoarcus* sp. CIB (4.57 × 10^4^ bacteria per g of roots) (Figure 3B). Therefore, the results obtained reveal that the inactivation of the *eps* cluster reduces the ability of *Azoarcus* sp. CIB to interact with the rice plant. These results provide strong experimental support on the role of the exopolysaccharide(s) promoting the endophytic lifestyle in members of the *Azoarcus*/*Aromatoleum* group.

### 3.3. c-di-GMP Levels Influence the Endophytic Lifestyle of Azoarcus sp. CIB

As mentioned in the Introduction, c-di-GMP is a second messenger involved in the regulation of many bacterial functions. However, its role in beneficial plant-bacteria interactions and, more specifically, in their endophytic relationships, is poorly studied and far to be understood. Since the production and/or functionality of *pili* type IV, flagellum and exopolysaccharide (all of which were shown above to be involved in the *Azoarcus* sp. CIB-rice interactions) have been related with the intracellular levels of c-di-GMP in bacteria [88,89,90], it was tempting to speculate that c-di-GMP could be also regulating the endophytic lifestyle of strain CIB. To study further the proposed new function of c-di-GMP in bacterial endophytes, we checked recombinant *Azoarcus* sp. CIB strains containing different intracellular levels of c-di-GMP. To this end, we used *Azoarcus* sp. CIB (pIZ2133), expressing the PA2133 c-di-GMP phosphodiesterase from *P. aeruginosa* PAO1 [26], and *Azoarcus* sp. CIB (pIZ4959), expressing the PP4959 diguanilate cyclase from *P. putida* KT2440 [91], two strains described for their reduced or increased levels of c-di-GMP, respectively, with respect to those of the parental strain *Azoarcus* sp. CIB (pIZ1016) ([68], Table 1). To confirm that the two recombinant strains showed the phenotypes anticipated according to the variations in c-di-GMP levels [28], we analyzed first their swarming and swimming motility. As expected, whereas *Azoarcus* sp. CIB (pIZ2133) (reduced c-di-GMP levels) showed an increased swarming and swimming motility, the strain *Azoarcus* sp. CIB (pIZ4959) (increased c-di-GMP levels) showed a reduced swarming and swimming motility with respect to that of the control strain (Figure 4). The growth rates of the CIB control strain and its derivative strains CIB (pIZ2133) and CIB (pIZ4959) were similar (data not shown), so the observed different motility cannot be attributed to differences in their cell growth. The morphology of the colonies was later analyzed and whereas *Azoarcus* sp. CIB (pIZ2133) generated smoother and more plain colonies than the control strain, *Azoarcus* sp. CIB (pIZ4959) showed rough colonies (Figure 4), which is consistent with a higher exopolysaccharide synthesis [28].

To analyze whether the altered levels of c-di-GMP in *Azoarcus* sp. CIB (pIZ2133) and *Azoarcus* sp. CIB (pIZ4959) strains might influence their plant colonization abilities, rice seedlings colonization tests were performed. As shown in Figure 5, the number of endophytes recovered from seedlings inoculated with *Azoarcus* sp. CIB (pIZ2133) cells (1.46 × 10^3^ bacteria per g of root), was more than one order of magnitude lower than the number of endophytes recovered after inoculation with the CIB control strain (2.5 × 10^4^ bacteria per g of root). On the other hand, inoculation with the strain *Azoarcus* sp. CIB (pIZ4959) led to 7.63 × 10^3^ endophytes per g of root, which represents a loss of efficiency in colonization of 70% with respect to the control strain. Therefore, these data suggest that either the artificial decrease or increase of c-di-GMP levels in *Azoarcus* sp. CIB affect negatively the endophytic colonization of rice by this bacterium. These results are in agreement with previous observations that any modification in the homeostasis of c-di-GMP negatively affects the interactions between the plant and pathogenic or symbiont bacteria [41,42,43]. These results can be explained by taken into account that both motility and adhesion are needed for successful plant colonization [2]. Thus, an increase in c-di-GMP levels leads to greater adhesion but, at the same time, to a decrease in bacterial motility; conversely, a decrease of c-di-GMP levels enhance bacterial motility but are detrimental for adhesion and formation of microcolonies needed for the entrance of the bacteria to the internal tissues of the plant [2,92].

To try to determine if the presumed modification of c-di-GMP levels in *Azoarcus* sp. CIB affected the transcription of genes involved in bacterial motility and/or exopolysaccharide formation, an expression analysis of the *pilY1* gene (encodes a protein for the synthesis and stabilization of *pili* type IV), *fliC* gene (encodes a structural protein of the flagellum) and the *epsF* gene (encodes a protein for exopolysaccharide modification) was performed. The results obtained revealed that the expression of the *epsF* gene was not significantly altered in any of the two strains that have modified levels of c-di-GMP, i.e., *Azoarcus* sp. CIB (pIZ2133) and *Azoarcus* sp. CIB (pIZ4959) (Figure 6). These results suggest that c-di-GMP does not control the exopolysaccharide genes at the transcriptional level, pointing to a post-transcriptional control, as already described for exopolysaccharide genes in other bacteria [93], as a more likely regulatory mechanism. Similarly, the expression of the *pilY1* and *fliC* genes was not significantly altered in *Azoarcus* sp. CIB (pIZ2133) (reduced c-di-GMP levels) when compared with their expression in the control strain (Figure 6), suggesting that the increased motility observed in the former strain might be controlled by a c-di-GMP dependent post-transcriptional regulation as already shown in *P. aeruginosa* PAO1 [94]. In contrast, the expression of *pilY1* and *fliC* genes was 3–3.5 times lower in the strain *Azoarcus* sp. CIB (pIZ4959) than in the control strain (Figure 6), suggesting that the increase of c-di-GMP levels leads to a transcriptional repression of such genes. The repression of the *pil* and *fli* genes in response to elevated cellular levels of c-di-GMP was previously observed in other bacteria such as *V. cholerae* [95,96], and is in agreement with the reduced motility observed in swarming and swimming assays (Figure 4). It is known that the c-di-GMP mediated transcriptional regulation of motility genes takes place through the FleQ regulator in *P. aeruginosa* and *P. putida* [37,97,98], or through alterations in chemotactic machinery affecting receptors with PilZ domains [99,100]. Since *Azoarcus* sp. CIB does not have a FleQ ortholog and only a protein with a putative PilZ domain has been identified by genome search, it is currently difficult to predict the signaling cascade involved in the transcriptional control mediated by c-di-GMP on the *pil* and *fli* genes in the CIB strain.

In summary, all the results presented here suggested for the first time how the modification of intracellular levels of c-di-GMP in an endophytic bacterium affects its ability to colonize the host plant, and they reveal an unnoticed role of c-di-GMP in bacteria controlling the endophytic lifestyle. It is tempting to speculate that, as described in the endophyte bacterium *Azoarcus* sp. CIB, the modification of c-di-GMP levels may modulate pathogenesis of plant pathogenic bacteria. In this sense, there have been described some examples where changes in the levels of c-di-GMP control pathogenesis, e.g., modulating flagellar swimming motility and plant disease progression in *Erwinia amylovora* [101] or *Pseudomonas syringae* pv. tomato DC3000 [102]. However, these findings could not be generalized to all pathogenic bacteria. Thus, increasing c-di-GMP levels in *P. savastanoi* pv. savastanoi might reduce the necrosis observed on tomato or bean but the development of other disease symptoms did not seem significantly affected by high c-di-GMP [45]. More work needs to be done to elucidate the signaling cascade that responds to the levels of c-di-GMP in *Azoarcus* sp. CIB.

## 4. Conclusions

The results of the present study describe the participation of the motility mediated by the flagellum and *pili* type IV, the adhesion modulated by exopolysaccharides and, remarkably, the unnoticed participation of the intracellular levels of c-di-GMP in the colonization of rice roots by *Azoarcus* sp. CIB. More experiments need to be developed to understand the complete landscape of molecular determinants involved in the signaling cascade that responds to the levels of c-di-GMP in *Azoarcus* sp. The results of these experiments might be used to set strategies to modulate the interaction of endophytes with their target plant hosts.

## Figures and Tables

**Figure 1 microorganisms-09-00554-f001:**
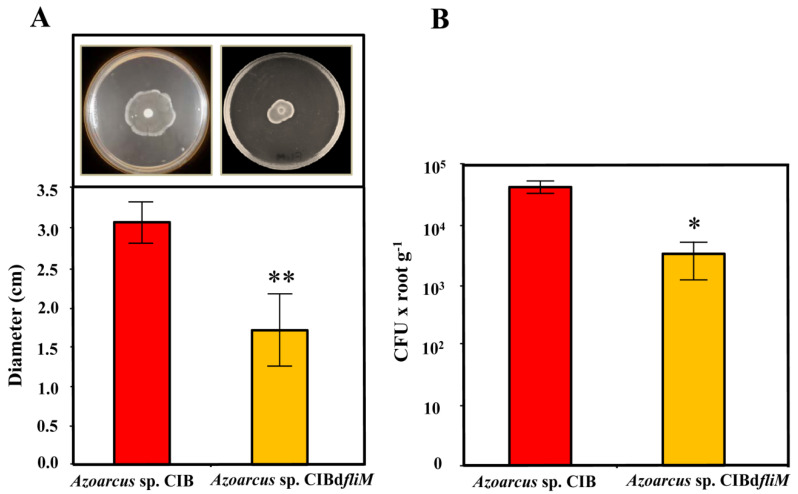
Swimming and rice colonization phenotypes of the *Azoarcus* sp. CIB and *Azoarcus* sp. CIBd*fliM* strains. (**A**) Swimming motility of the *Azoarcus* sp. CIB (carrying the control plasmid pSEVA237) and *Azoarcus* sp. CIBd*fliM* strains, and quantification of the diameter of the halos (*n* = 3); the error bars indicate the standard deviation. Two asterisks mean a very significant difference (*p* = 0.0076), according to paired *t*-test. (**B**) Quantification of the number of endophytes recovered from rice roots after inoculation with *Azoarcus* sp. CIB (pSEVA237) and *Azoarcus* sp. CIBd*fliM* strains. Graph shows the CFU values per gram of root (fresh weight) from three independent experiments ± standard deviation. An asterisk means a significant difference (*p* = 0.0319), according to paired *t*-test.

**Figure 2 microorganisms-09-00554-f002:**
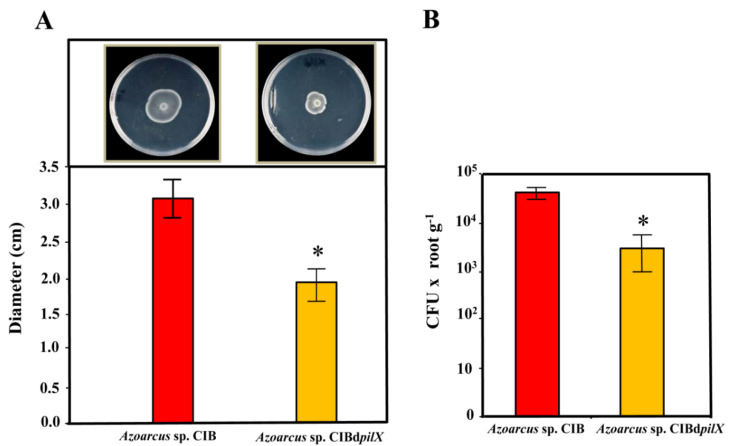
Motility and rice colonization phenotypes of the *Azoarcus* sp. CIB and *Azoarcus* sp. CIBd*pilX* strains. (**A**) Swarming motility of *Azoarcus* sp. CIB (carrying the control plasmid pSEVA237) (left) and *Azoarcus* sp. CIBd*pilX* (right), and quantification of the diameter of the halos (*n* = 3); the error bars indicate the standard deviation. An asterisk means a significant difference (*p* = 0.0377), according to paired *t*-test. (**B**) Quantification of the number of endophytes recovered from rice roots after inoculation of rice seedlings with the *Azoarcus* sp. CIB (pSEVA237) and *Azoarcus* sp. CIBd*pilX*. The graph shows the CFU values per gram of root (fresh weight) of three independent experiments; error bars indicate standard deviation. An asterisk means a significant difference (*p* = 0.0219), according to paired *t*-test.

**Figure 3 microorganisms-09-00554-f003:**
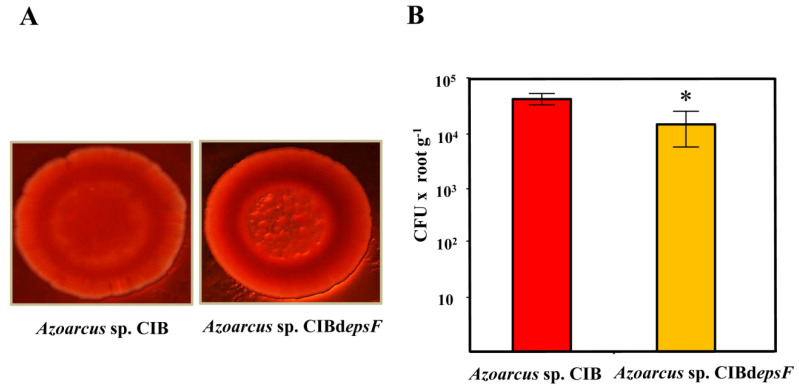
Colony morphology and rice colonization capacity of the *Azoarcus* sp. CIB and *Azoarcus* sp. CIBd*eps*F strains. (**A**) Morphology of the colonies of *Azoarcus* sp. CIB (pSEVA237) and *Azoarcus* sp. CIBd*epsF* strain grown in VM solid medium supplemented with Congo red (0.5 g L^−1^). The colonies were visualized with a Leica MZ16FA stereomicroscope. (**B**) Quantification of the number of endophytes recovered from rice roots after inoculation with *Azoarcus* sp. CIB (pSEVA237) and *Azoarcus* sp. CIBd*epsF* strains. Graph shows the CFU values per gram of root (fresh weight) of three independent experiments ± standard deviation. An asterisk means a significant difference (*p* = 0.0375), according to paired *t*-test.

**Figure 4 microorganisms-09-00554-f004:**
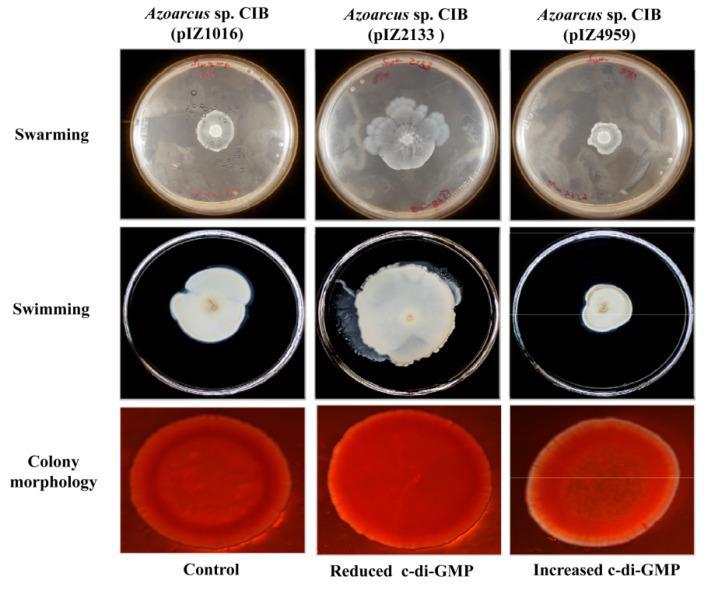
Motility and morphology of the colonies of *Azoarcus* sp. CIB strains expressing genes involved in the metabolism of c-di-GMP. The plates were inoculated with *Azoarcus* sp. CIB (pIZ1016) (empty plasmid), *Azoarcus* sp. CIB (pIZ2133) (expressing the PA2133 phosphodiesterase) and *Azoarcus* sp. CIB (pIZ4959) (expressing the PP4959 diguanylate cyclase).

**Figure 5 microorganisms-09-00554-f005:**
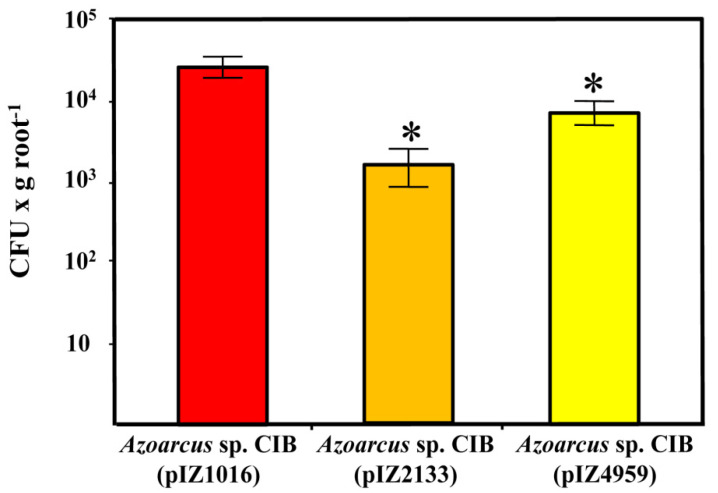
Rice roots colonization by *Azoarcus* sp. CIB strains producing different levels of c-di-GMP. Rice seedlings were inoculated with *Azoarcus* sp. CIB (pIZ1016) (empty plasmid), *Azoarcus* sp. CIB (pIZ2133) (expressing the PA2133 phosphodiesterase) and *Azoarcus* sp. CIB (pIZ4959) (expressing the PP4959 diguanylate cyclase). Plants were grown at 25 °C for 5 days and bacteria present inside the roots were determined as described in Methods. The graph shows the CFU values per gram of root (fresh weight) of three independent experiments; error bars represent the standard deviation. Values of *Azoarcus* sp. CIB (pIZ1016) versus *Azoarcus* sp. CIB (pIZ2133) (*p* = 0.0142) and *Azoarcus* sp. CIB (pIZ4959) (*p* = 0.0154) were significantly different from each other and are indicated with one asterisk (paired *t*-test analysis).

**Figure 6 microorganisms-09-00554-f006:**
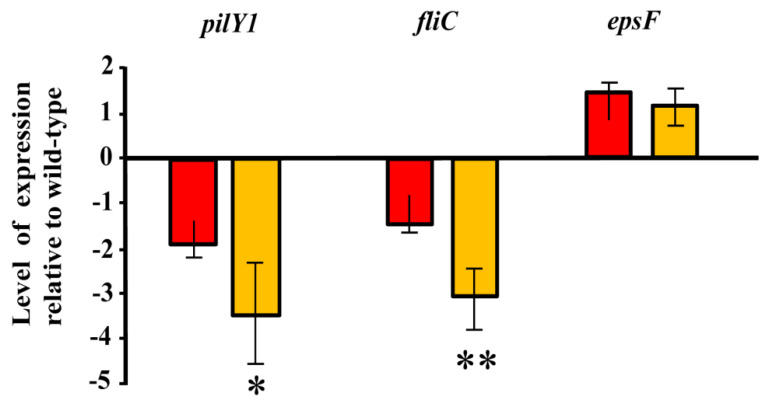
Expression of the *pilY1*, *fliC* and *epsF* genes in *Azoarcus* sp. CIB strains producing different levels of c-di-GMP. The transcript levels of the *pilY1*, *fliC* and *epsF* genes in the *Azoarcus* sp. CIB (pIZ2133) (**red columns**) and *Azoarcus* sp. CIB (pIZ4959) (**yellow columns**) strains grown in VM-ethanol medium for 24 h, were determined by quantitative RT-PCR using *dnaE* gene as internal control, as detailed in Methods. The expression of the genes in each strain is shown relative to their expression in the control strain *Azoarcus* sp. CIB (pIZ1016). The mean value and the standard deviation corresponding to the results of three independent experiments are indicated. Values of expression of *pilY* gene (*p* = 0.0174) in *Azoarcus* sp. CIB (pIZ2133) and *Azoarcus* sp. CIB (pIZ4959), were significantly different (*p* = 0.0174; one asterisk). Values of expression of *fliC* gene in *Azoarcus* sp. CIB (pIZ2133) and *Azoarcus* sp. CIB (pIZ4959), were very significantly different (*p* = 0.0095; two asterisks). Values of expression of *epsF* gene in *Azoarcus* sp. CIB (pIZ2133) and *Azoarcus* sp. CIB (pIZ4959) were not significantly different (*p* = 0.0516). All the statistic data were obtained according to unpaired *t*-test.

**Table 1 microorganisms-09-00554-t001:** Bacterial strains and plasmids used in this study.

Strain or Plasmid	Relevant Genotype and Characteristic(s)	Reference or Source
*E. coli* strains		
DH10B	F’, *mcrA*, *Δ(mrr hsdRMS-mcrBC*), *Φ80lacZΔM15*,*ΔlacX74*, *deoR*, *recA1*, *araD139*, *Δ(ara-leu)7697*, *galU*, *galK*, *rpsL* (Sm^r^), *endA1*, *nupG*,	Life Technologies
S17-1λ*pir*	Tp^r^ Sm^r^ *recA thi hsdRM^+^* RP4::2-Tc::Mu::Km, λ*pir* phage lysogen	[64]
CC118	*Δ(ara-leu)*, *araD*, *ΔlacX7*, *galE*, *galK*, *phoA20*, *rpoB*, *thi-1*, *rpsE*, (Sp^r^), (Rf^r^), *argE*, (Am), *recA1*	[65]
*Azoarcus* strains		
CIB	Wild type strain	[59]
CIBd*pilX*	Km^R^, CIB mutant by insertion in the *pilX* gene	This work
CIBd*fliM*	Km^R^, CIB mutant by insertion in the *fliM* gene	This work
CIBd*epsF*	Km^R^, CIB mutant by insertion in the *epsF* gene	This work
Plasmids		
pSEVA237	Km^r^, *ori* pBBR1, harbors the *gfp* gene under the control of the *PlexA* promoter	[66]
pIZ1016	Gm^r^, *ori* pBBR1MCS-5 derivative vector for cloning and expression harboring the *Ptac* promoter and the *lacI* gene	[67]
pIZ2133	Gm^R^, pIZ1016 derivative containing the gene PA2133 from *P. aeruginosa* PAO1 under the control of the *Ptac* promoter	[68]
pIZ4959	Gm^R^, pIZ1016 derivative containing the gene PP4959 from *P. putida* KT2440 under the control of the *Ptac* promoter	[68]
pK18*mob*	Km^R^, *ori*ColE1, Mob^+^, *lac*Zα, suicide vector for the construction of insertion mutants by homologous recombination	[69]
pK18*mobpilX*	Km^R^, pK18*mob* derivative that includes a 455 bp *pilX Hin*dIII/*Bam*HI internal fragment	This work
pK18*mobfliM*	Km^R^, pK18*mob* derivative that includes a 512 bp *fliM Hin*dIII/*Bam*HI internal fragment	This work
pK18*mobepsF*	Km^R^, pK18*mob* derivative that includes a 635 bp *epsF Hin*dIII/*Bam*HI internal fragment	This work

Km^r^: kanamycin resistant; Gm^r^: gentamicin resistant; Sm^r^: streptomycin resistant.

**Table 2 microorganisms-09-00554-t002:** Composition of growth media used in this study.

Medium	Quantity per Litre of Distillated Water
**MA (pH 7.5)**	
KH_2_PO_4_	0.33 g
Na_2_HPO_4_	1.20 g
NH_4_Cl	0.11 g
MgSO_4_ × 7H_2_O	0.10 g
CaCl_2_	0.04 g
Trace elements (stock solution 100×) (pH 6.5)	
Nitrilotriacetic acid	1.50 g
MgSO_4_ × 7H_2_O	3.00 g
MnSO_4_ × 2H_2_O	0.50 g
NaCl	1.00 g
FeSO_4 ×_ 7H_2_O	0.10 g
CoSO_4_ × 7H_2_O	0.18 g
CaCl_2_ × 2H_2_O	0.10 g
ZnSO_4_ × 7H_2_O	0.18 g
CuSO_4_ × 5H_2_O	0.01 g
KAl(SO_4_)_2_ × 12H_2_O	0.02 g
H_3_BO_3_	0.01 g
Na_2_MoO × 2H_2_O	0.01 g
NiCl_2_ × 6H_2_O	0.025 g
Na_2_ScO_3_ × 5H_2_O	0.30 mg
Vitamin solution (stock 1000×)	
Biotin	20 mg
Folic acid	20 mg
Pyridoxine-HCl	10 mg
Thiamine-HCl × 2H_2_O	50 mg
Riboflavin	50 mg
Nicotinic acid	50 mg
Calcium *D*-pantothenic acid	50 mg
Vitamin B12	50 mg
*p*-aminobenzoic acid	50 mg
Modified VM-ethanol rich medium (pH 6.8)	
KH_2_PO_4_	0.40 g
K_2_HPO_4_	0.60 g
NaCl	1.10 g
NH_4_Cl	0.50 g
MgSO_4_ × 7H_2_O	0.20 g
CaCl_2_	26 mg
MnSO_4_	10 mg
Na_2_MoO_4_	2 mg
Fe(III)-EDTA	66 mg
Yeast extract	1.00 g
Bactopeptone	3.00 g
Ethanol	6.00 mL

## Data Availability

The data presented in this study are available within this article.

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
