# Peer review of "Motility, Adhesion and c-di-GMP Influence the Endophytic Colonization of Rice by Azoarcus sp. CIB"

_microorganisms, 2021, doi:10.3390/microorganisms9030554_

Round 1

Reviewer 1 Report

The manuscript "Motility, adhesion and c-di-GMP influence the endophytic colonization 2 of rice by Azoarcus sp. CIB" presents interesting data regarding plant-microbe interaction mechanisms of a select beneficial bacterial strain with an important crop such as rice.

The results presented can be relevant to advance the knowledge of plant-microbe interaction both with beneficial and pathogenic bacteria.

However, there are some issues that should be addressed:

  • The major concern with the study is the lack of controls regarding the mutants used in the research. For the part of the c-di-GMP, the authors correctly used a wild-type strain, an overproducer, and a mutant with impaired production, providing convincing evidence that the amount of molecule that was produced, as the two mutants had opposite phenotypes and the wild-type had an intermediate one. For the deletion mutants used in for the flagellum and pili formation, the evidence is sadly not solid. Constructing mutants that complement the deletion and restore wild-type function would be necessary to demonstrate that it's the intended deletion that causes the observed phenotype. Also, providing quantitative PCR data for the target genes in all mutants could help prove the point that the deletion/expression of target gene was successful.
  • I would advise the authors to provide the definition of "endophytes" that they use throughout the study. It seems that they differentiate between the terms endophyte, symbiont, and pathogen, but no definition is given for what they mean with the two terms. While the last one ,'pathogen', is self-explanatory, several definitions exist for both 'endophyte' and 'symbiont'.

- The statistical analysis performed is not only not appropriate for the data provided, but is impossible to perform. In most of the presented pictures the authors state that they used One-Way ANOVA followed by Bonferroni post-hoc test. Yet they have only 2 "groups" to compare (generally wild-type versus mutant). With that kind of dataset, SPSS will give an error message regarding the post-hoc test, which CANNOT be carried out without having 3 or more groups. The authors should provide the actual method used to carry out statistical analyses.
Also, for figures 5 and 6, it seems from the caption that rather than doing a comparison between all three groups, the authors perform a separate pairwise comparison between each mutant and the wild-type strain. While this result can be obtained from a One-Way ANOVA followed by post-hoc test, if such an analysis was carried out it would be more correct to indicate also if there is difference between the two mutants, actually showing all the results of the analysis, not just the comparisons with the wild type, that make it seem like the results of two t-tests, rather than an ANOVA.

Author Response

The manuscript "Motility, adhesion and c-di-GMP influence the endophytic colonization 2 of rice by Azoarcus sp. CIB" presents interesting data regarding plant-microbe interaction mechanisms of a select beneficial bacterial strain with an important crop such as rice.

The results presented can be relevant to advance the knowledge of plant-microbe interaction both with beneficial and pathogenic bacteria.

 However, there are some issues that should be addressed:

Q1. The major concern with the study is the lack of controls regarding the mutants used in the research. For the part of the c-di-GMP, the authors correctly used a wild-type strain, an overproducer, and a mutant with impaired production, providing convincing evidence that the amount of molecule that was produced, as the two mutants had opposite phenotypes and the wild-type had an intermediate one. For the deletion mutants used in for the flagellum and pili formation, the evidence is sadly not solid. Constructing mutants that complement the deletion and restore wild-type function would be necessary to demonstrate that it's the intended deletion that causes the observed phenotype. Also, providing quantitative PCR data for the target genes in all mutants could help prove the point that the deletion/expression of target gene was successful.

A1. Thank you for the comment regarding the complementation tests. We should remind that we used a technique of gene disruption in Azoarcus sp. CIB (successfully applied in the last 15 years in our laboratory) through single homologous recombination. This technique usually generates polar effects downstream of the mutated gene, as shown previously (see for instance López-Barragán et al., 2004). This information has been incorporated in the new version (Material and Methods section 2.4.). Thus, complementation of a disrupted gene within an operon might not be achieved by just expressing a wild-type copy of the disrupted gene in a plasmid.

We present  experiments demonstrating that the mutants were accurately constructed by the analysis of the expression of the mutated genes by RT-PCR (Fig. S1). The oligonucleotides used in the RT-PCRs have been also detailed in Table S2.

Although the results obtained with our mutant strains do not allow us to conclude that the disrupted gene is the only one responsible for the observed phenotype, they can be used to confirm the role of such operon in the particular phenotype, which in fact was what we wanted to demonstrate.

Q2. I would advise the authors to provide the definition of "endophytes" that they use throughout the study. It seems that they differentiate between the terms endophyte, symbiont, and pathogen, but no definition is given for what they mean with the two terms. While the last one,'pathogen', is self-explanatory, several definitions exist for both 'endophyte' and 'symbiont'.

A2. OK, a definition of endophyte has been incorporated in the Introduction of the new version. The term “symbiont” has been substituted by “plant-associated bacteria” to avoid misinterpretations.

Q3. The statistical analysis performed is not only not appropriate for the data provided, but is impossible to perform. In most of the presented pictures the authors state that they used One-Way ANOVA followed by Bonferroni post-hoc test. Yet they have only 2 "groups" to compare (generally wild-type versus mutant). With that kind of dataset, SPSS will give an error message regarding the post-hoc test, which CANNOT be carried out without having 3 or more groups. The authors should provide the actual method used to carry out statistical analyses.
Also, for figures 5 and 6, it seems from the caption that rather than doing a comparison between all three groups, the authors perform a separate pairwise comparison between each mutant and the wild-type strain. While this result can be obtained from a One-Way ANOVA followed by post-hoc test, if such an analysis was carried out it would be more correct to indicate also if there is difference between the two mutants, actually showing all the results of the analysis, not just the comparisons with the wild type, that make it seem like the results of two t-tests, rather than an ANOVA.

Q3. Thank you very much for the comment. The reviewer is right. Statistical analysis has been revised and now the t-test has been used with all the data presented in the manuscript. The information about the program used in the statistical analysis has been included in Material and Methods (section 2.11.). Also the P values from the t-test performed are displayed in the figure legends, indicating the samples that were compared in each case.

Reviewer 2 Report

The manuscript entitled “Motility, adhesion and c-di-GMP influence the endophytic colonization of rice by Azoarcus sp. CIB” have explored the valuable information on the genetic determinants involved in the colonization of plants by endophytic bacteria. The paper has been well written and can be recommended for publication. However, prior to final acceptance, the author need to revise the manuscript and should address the following comments:

Comments:

  1. Is the strain Azoarcus sp. CIB being newly isolated or purchased? If this strain was isolated by author the, please include the methods for isolation.
  2. As mentioned in the introduction, are there any assessment done by the author to assure the aromatic compound degradation and metal tolerance?
  3. In section 2.2: it is better to provide the information of media in the tabulated form.
  4. A detail qRT-PCR condition should be provided as supplementary file.

Author Response

The manuscript entitled “Motility, adhesion and c-di-GMP influence the endophytic colonization of rice by Azoarcus sp. CIB” have explored the valuable information on the genetic determinants involved in the colonization of plants by endophytic bacteria. The paper has been well written and can be recommended for publication. However, prior to final acceptance, the author need to revise the manuscript and should address the following comments:

Comments:

Q1. Is the strain Azoarcus sp. CIB being newly isolated or purchased? If this strain was isolated by the author, please include the methods for isolation.

A1. The CIB strain is not a new isolate. This strain was isolated 20 years ago as already described in López-Barragán et al. (2004) (reference #59). The strain was then deposited in the Spanish Type Culture Collection (CECT #5669) as stated in Material and Methods section 2.1.

Q2. As mentioned in the introduction, are there any assessment done by the author to assure the aromatic compound degradation and metal tolerance?

A2. We have been working with the CIB strain for the last two decades. We have already experimentally demonstrated and published the ability to degrade aromatics aerobically and anaerobically. Also we have described the genes that might be involved in metal tolerance and demonstrated experimentally the ability to tolerate arsenic, selenite, cadmium and tellurium. In the Introduction we include the references of some of the publications that deal with these skills (#59-62).

Q3. In section 2.2: it is better to provide the information of media in the tabulated form.

A3. Ok, the media composition is presented in the new Table 2.

Q4. A detail qRT-PCR condition should be provided as supplementary file.

A4. Thank you. We have incorporated this information in the section 2.5. of Material and Methods. Although we do not consider necessary to open a new text in the supplementary file, if the reviewer or the editor consider that the text is too explicit for Material and Methods, we can move it to the supplementary file.  

Round 2

Reviewer 1 Report

The corrections and clarifications made by the authors, together with the added supplementary materials, are sufficient and adequate to answer to my comments. I have no further points of critique for this study.

This manuscript is a resubmission of an earlier submission. The following is a list of the peer review reports and author responses from that submission.

Round 1

Reviewer 1 Report

The manuscript presents interesting experiments regarding the dissection of some mechanic determinants of plant-bacteria interaction, using a beneficial microbe and a staple food model plant, rice.

While the hypothesis behind the study is interesting and the authors do work in the right direction towards demonstrating this hypothesis, the study lacks some important trails that undermine its overall soundness.

First of all, the study would benefit from the use of double mutants (for example strains impaired in both fliM and pilX synthesis), and must produce revertant mutants to have definitive proof that the intended mutation is the one and only determinant of the observed phenotype. While the gene expression assays carried out do certify that the target genes are affected by the mutations, they do not exclude the possibility of additional, unintended perturbations in the genome being behind this effect.

Furthermore, as far as I could see, the authors did not actually evaluate the c-di-GMP values in the mutants in any direct way. Therefore, while it is extremely logical and likely that the mutations that they caused had the desired effect of increasing or reducing the quantity of this molecule in the affected strains, all the sentences that speak about the levels or concentration of c-di-GMP (including the title) must either be mitigated, saying explicitly that it is a deduction, or the level of the molecule must actually be quantified to support these conclusions.

Reviewer 2 Report

I read the manuscript by Fernández-Llamosas et al. where they described their works on identifying mechanisms of plant-endophyte colonization. Any aspects of studying endophyte is important given their perceived significance and yet many unknown characteristics. Of particular interest is the molecular mechanisms that modulate their attachement with host and subsequent colonization and mutualisms. Most previous studies have been on plant pathogenic bacteria and we have decent amount of information about their interaction with plant.

This paper describes the roles of genes responsible for motility and exopolysaccharides modulation and more importantly the involvement of the bacterial second messenger c-di-GMP in the colonization of a Azoarcus sp. CIB in rice roots.

In general, the introduction and materials and methods are written well. However, some sentences are two long to understand. For example, L41-45, L73-77 etc.  Also, it is need to improve English. Sometimes it is hard to understand the intended meaning. For example, in L33 “Hardoim et al. …….able to interact with plants.”  Also, L 26-32 and L69. In methodology, I did not find the number of replications. In the figure, they mentioned n=3 which is not very clear. The presentation with bargraph does not seems to be very appropriate for a good scientific journal which have been discussed a lot elsewhere. For mutation experiment, did author performed any complementation test?

As mentioned by the author, results in 3.1 and 3.2 have also been described in another endophyte of same species but in different strains Azoarcus sp. BH72. In this respect, this is not something a novel finding. Could author state like this “L292- These results reveal an important role of the flagellum at some stage of the endophytic colonization of rice by Azoarcus sp. CIB’. Author did not mention if Azoarcus sp. BH72 has also been studied on rice or another host. If on rice, what did author expect from similar study. Please discuss more.

The roles and involvement c-di-GMP levels on Azoarcus sp. CIB colonization seems to be a novel finding on any endophyte colonization. It is not clear if the author is continuing the study further the mechanisms of the signaling cascade in plant in Azoarcus sp. CIB which would be a very interesting aspect. Author did not discuss further or proposed any speculation on pathways. I would suggest to check the downstream analyses for a deeper understanding of the plant-endophyte colonization and interaction. Also, a nice discussion is expected that highlight any difference from pathogenic vs endophytic colonization lifestyles and if the findings obtained by the author could be generalized (at least to draw a hypothesis/speculation) based on the currently available knowledge. Since, the author has only small number of novel findings without complementation mutation and proof of any post-transcriptional control, I hesitate it could be published as a full paper in this journal.  

Reviewer 3 Report

 The authors had studied a very interesting  aspect of endophytic colonization  in the manuscript "Motility, adhesion and c-di-GMP levels influence the endophytic colonization of rice by Azoarcus sp. CIB". The  experiment design and  their presentation is  well.